# Effects of Amylopectins from Five Different Sources on Disulfide Bond Formation in Alkali-Soluble Glutenin

**DOI:** 10.3390/foods12020414

**Published:** 2023-01-16

**Authors:** Yu Zhou, Jinjin Zhao, Junjie Guo, Xijun Lian, Huaiwen Wang

**Affiliations:** 1Tianjin Key Laboratory of Food Biotechnology, School of Biotechnology and Food Science, Tianjin University of Commerce, Tianjin 300134, China; 2School of Mechanical Engineering, Tianjin University of Commerce, Tianjin 300134, China

**Keywords:** alkali-soluble glutenin, amylopectin, ^13^C solid-state NMR, disulfide bond

## Abstract

Wheat, maize, cassava, mung bean and sweet potato starches have often been added to dough systems to improve their hardness. However, inconsistent effects of these starches on the dough quality have been reported, especially in refrigerated dough. The disulfide bond contents of alkali-soluble glutenin (ASG) have direct effects on the hardness of dough. In this paper, the disulfide bond contents of ASG were determined. ASG was mixed and retrograded with five kinds of amylopectins from the above-mentioned botanical sources, and a possible pathway of disulfide bond formation in ASGs by amylopectin addition was proposed through molecular weight, chain length distribution, FT-IR, ^13^C solid-state NMR and XRD analyses. The results showed that when wheat, maize, cassava, mung bean and sweet potato amylopectins were mixed with ASG, the disulfide bond contents of alkali-soluble glutenin increased from 0.04 to 0.31, 0.24, 0.08, 0.18 and 0.29 μmol/g, respectively. However, after cold storage, they changed to 0.55, 0.16, 0.26, 0.07 and 0.19 μmol/g, respectively. The addition of wheat amylopectin promoted the most significant disulfide bond formation of ASG. Hydroxyproline only existed in the wheat amylopectin, indicating that it had an important effect on the disulfide bond formation of ASG. Glutathione disulfides were present, as mung bean and sweet potato amylopectin were mixed with ASG, and they were reduced during cold storage. Positive/negative correlations between the peak intensity of the angles at 2*θ* = 20°/23° and the disulfide bond contents of ASG existed. The high content of hydroxyproline could be used as a marker for breeding high-quality wheat.

## 1. Introduction

Wheat gluten is a co-product of wheat starch production in China. It can increase the viscoelasticity of cooked wheat-based food at a low cost in the food industry [1]. According to its solubility in aqueous alcohol, two fractions of wheat gluten are isolated, of which the gliadins are soluble and the glutenins are insoluble. Gliadins are monomeric proteins, and glutenins are inter-chain disulfide-linked polymers [2]. These inter-chain disulfide bonds of glutenin ensure the stability of the three-dimensional gluten gelation network and provide functional properties to wheat flour products [3]. Dough formation is necessary for the preparation of most wheat flour products. Mixing flour and water leads to the formation of disulfide bonds, which serve a vital function in maintaining the structural and functional properties of gluten [4]. In the process of turning wheat flour into dough, those glutenins with inter-chain disulfide bonds consist of the backbone network of gluten. Some studies have found that intramolecular disulfide bonds also exist in glutenin polymers [5]. Glutenins are aggregated proteins with a high molecular weight (*M_w_*) distribution ranging from 10^5^ to 10^7^ g/mol [6]. Furthermore, the glutenins are categorized into two subunits: high-molecular-weight subunits of approximately 70,000 to 90,000 g/mol and low-molecular-weight subunits of approximately 30,000–40,000 g/mol [7], and the former are considered the most important determinant of the structure of these polymers [8]. Regarding these glutenins, Gianibelli et al. (2001) reported that their C-terminal domain, with a constant 42-amino-acid residue, contains one conservative cysteine, the and N-terminal domain, with an approximately 80- to 100-amino-acid residue, maintains two to five conservative cysteine residues [7]. It is reported that the elasticity of wheat gluten is mainly correlated with the β-sheets and β-turn structures of gluten [9]. Further studies have shown that the major elastic components of gluten are high-molecular-weight subunits of glutenin, in which the elastic β-sheet structure is formed at the cost of repetitive β-turns in the central domain [10]. It is well-known that low- and medium-gluten flours exhibit poor viscoelasticity in steamed bread and noodle processing in China, which is caused by a lower level of disulfide bond formation. A recent study [11] showed that blending gluten with wheat, corn, tapioca, sweet potato and potato starches has different effects on its disulfide bond formation, and wheat and potato starch–gluten dough shows the highest and lowest disulfide bond contents, respectively. The authors, in their previous research, found that the addition of wheat amylopectin enhances the disulfide bond formation of gliadin but addition of wheat amylose retards (data not shown). Such results agree well with the fact that the addition of a large amount of amylose reduces gluten formation in wheat flour, as confirmed in previous unpublished studies. Starch may serve as an accelerant in the formation of disulfide bonds by affecting the secondary structure of gluten. The structure of polymeric gluten is difficult to study because it consists of dozens of different peptides bound by disulfide bonds with molecular weights ranging from millions to hundreds of millions. Since inter-chain glutenin conferred gluten elasticity by disulfide bond formation, amylopectins from five different botanical sources were selected to investigate the effects of their addition on the disulfide bond formation of ASG (corresponding to high-molecular-weight glutenin molecules).

Chen et al. (2021) reported that higher temperatures increased sulfhydryl–disulfide bond (SH-SS) interchange to promote the aggregation of gluten [2], but whether it has the same effects on ASG needs to be studied further. Starch retrogradation essentially involves a reassociation of dispersed amylopectin into a more ordered structure stabilized by hydrogen bonds. It was found that hydrogen bonds produced in starch retrogradation also probably prevent the formation of the three-dimensional elastic network of alkali-soluble glutenin [12], thus producing certain effects on disulfide bond formation. The retrogradation of the disulfide bond formation of an amylopectin alkali-soluble glutenin mixture was also studied in the paper.

In this paper, the effects of mixing different amylopectins from wheat, maize, mung bean, tapioca and sweet potato starches with alkali-soluble glutenin on disulfide bond formation before and after retrogradation were investigated. The objective of the present study was to determine which amylopectin promotes the most significant glutenin disulfide bond formation and to deduce the possible interaction mechanism by comparing the results of molecular weight, chain length distribution, ^13^C solid-state NMR, IR and X-ray diffraction analyses.

## 2. Materials and Methods

### 2.1. Materials

Wheat and maize starches were purchased from He Nan Enmiao Food Co., Ltd. (Zhengzhou, China) sweet potato starch was purchased from the Beijing Gusong Economic and Trade Company (Beijing, China), mung bean starch was purchased from Hengshui Fuqiao Starch Co., Ltd. (Hengshui, China) and cassava starch was purchased from Guangxi Napoheshan Starch Co., Ltd. (Baise, China). Bacillus subtilis thermostable and mid-temperature α-amylase (12000 U/mL), microbial lipase (20,000 U/g) and neutral protease (50,000 U/g) were all produced by Beijing Solarbio Science & Technology Co. Ltd. (Beijing, China). Sodium hydroxide, sodium chloride and hydrochloric acid were obtained from Tianjin Fengchuan Chemical Reagents Co., Ltd. (Tianjin, China). Sodium chloride (Analytical reagent with 99.5% NaCl) was purchased from Tianjin Hengxing Chemical Reagent Manufacturing Co., Ltd. (Tianjin, China). Pullulanase (CAS No.: 9075-68-7, 1000 U/mL) was obtained from Beijing Solarbio Science & Technology Co., Ltd. The dialysis bags (viskase MD44-14, Avg. flat width 44 mm (diameter 28 mm), MWCO 14,000 Da) were produced by the Union Carbide Corporation (Danbury, Connecticut, United States). Tris, glycine, disodium EDTA, urea, guanidine hydrochloride and DTNB reagents were all obtained from Beijing Soleibo Science & Technology Co. Ltd. The P120H Ultrasonic Cleaner was produced by Elma Schmidbauer GmbH in Germany.

### 2.2. Methods

#### 2.2.1. Preparation and Isolation of the Different Amylopectins

Crude amylopectin was produced according to basic principle in the literature [13,14], according to which amylose can dissolve in 0.5% NaCl solutions but amylopectin cannot. In order to remove all the intact granules in the different starches, the granules were dipped and caused to swell in water at the gelation temperature, and then the swollen granules were fractured by the growth of ice crystals. Only amylose and amylopectin were present in the thawed starch solutions, and the crude amylopectins were isolated based on the basic principle in the literature [13,14]. After 1000 mL of distilled water was added, the wheat, maize, mung bean, tapioca and sweet potato starches, at the weight of 100g, respectively, were left to swell for 2h at 65 °C under continuous stirring until they were sticky. After cooling, the sticky starches were frozen at −18 °C in the freezer for one night. The frozen sticky starches were thawed at room temperature for 6h and centrifuged (3500× *g* for 3 min) to obtain the precipitates. Those precipitates were dissolved in 1% sodium chloride solution under constant stirring for 10 min to dissolve the amylose, and the precipitation (crude amylopectin) was obtained by centrifugation (4000× *g* for 5 min), and the supernatant fraction was discarded. Crude amylopectin (21~26 g) was isolated by repeated dissolution in 1% NaCl solution and centrifuged several times, as mentioned above, until no blue color appeared in the precipitation–iodine complex.

#### 2.2.2. The Purification of the Different Amylopectins

In order to completely remove a small amount of co-extracted or associated compounds, such as protein or lipid, from the crude amylopectins, the crude amylopectins were hydrolyzed by lipase and alkali protease in sequence, according to [15]. After being mixed with 100 mL deionized water, the different amylopectins (30 g wet weight, corresponding to ~8g dry weight) were hydrolyzed by lipase (20,000 U/g, 10 mg) for 24 h at 40 °C, and no measures were taken to terminate the enzyme activity. Then, the alkali protease (20,000 U/g, 10 mg) was used to hydrolyze the protein in the different amylopectins for 24 h at 55 °C after the pH of the above-mentioned solutions was adjusted to 8.0 with 2.0 mol·L^−1^ NaOH. Finally, both the lipase and alkali proteases were inactivated by boiling for 10 min. The solutions were centrifuged (4000× *g* for 10 min) to obtain the precipitates. The purified amylopectins (23~26 g wet weight, corresponding to 6~7 g dry weight) were prepared by washing the precipitates using deionized water several times until there was no turbidity when a drop in AgNO_3_ was added. The amylopectins for the IR, ^13^C solid-state NMR and X-ray diffraction analyses were obtained after the samples had been dried at 60 °C in an oven to obtain a constant weight.

#### 2.2.3. Preparation of the ASG

Glutenin was separated from gluten using a reference method, with certain modification [16]. Furthermore, based on [17], the parameters (solid/liquid = 1:25, 0.1% NaOH, while the extraction temperature and time were 30 °C and 120 min, and the temperature and time for condensation were 50 °C and 12 h) for the preparation of ASG are established through many pre-experiments. First, gluten (100g) was mixed with 65% ethanol at the ratio of 1:30 (g/v), and gliadin was dissolved in the ethanol solution under constant stirring for 2 h at 40 °C. Then, the precipitate was obtained by centrifugation at 4000× *g* for 10 min. Again, the gliadin in the precipitate was extracted by above-mentioned method several times until no viscous gliadin was clearly present. Finally, the glutenin, at the weight of 114 g (wet weight)/49 g (dry weight), was obtained by centrifugation. Approximately 20 g of wet glutenin (8.5 g dry weight) was added to 200 mL of 0.1% NaOH and stirred to extract the ASG for 120 min at room temperature. Then, the solution was centrifugated at 4000× *g* for 5 min to remove the alkali-insoluble glutenin. The above-mentioned supernatant was gently placed in beaker to obtain coagulation precipitates at 50 °C for 12 h. The coagulation precipitation was isolated by centrifugation at 4000× *g* for 5 min. This precipitation was dialyzed to remove the sodium and hydroxide ions, and 21.5 g wet/2.2 g dry weight of ASG was obtained.

#### 2.2.4. Mixture of the Different Amylopectins and ASG

The wheat, maize, potato, mung bean and cassava starches, at a wet weight of 1g, corresponding to dry weights of 0.090, 0.062, 0.075, 0.205 and 0.135 g, respectively, were mixed with ASG at a wet weight of 0.3 g (0.03g dry weight). The mixtures were stirred with a small bamboo skewer at 37 °C for 2 h. The wet mixtures were used to determine the contents of disulfide bonds, and the samples for IR, ^13^C solid-state NMR and X-ray diffraction were dried to a constant weight at 60 °C in an oven.

#### 2.2.5. Retrogradation of the Different Amylopectin–ASG Mixtures

The wet mixtures mentioned in Section 2.2.3, including the control group, were firstly gelatinized for 20 min at 95 °C by continuous stirring, and the gelatinized samples were autoclaved at 120 °C for 30 min. After that, they were retrograded at 4 °C for 7 d. The wet mixtures were used to determine the contents of disulfide bonds, and the samples for IR, ^3^C solid-state ^1^NMR and X-ray diffraction were dried to a constant weight at 60 °C in an oven.

#### 2.2.6. Determination of the Disulfide Bonds

The disulfide bond contents were determined according to the method described by Zhu et al. (2019) [18], with some modifications [19]. First, three solutions marked as A, B and C were prepared, as follows: solution A: Tris-Gly (pH 8.0), 1.0418 g Tris, 0.6756 g glycine and 0.1489 g EDTA were solubilized in 80 mL deionized water, where the volume was fixed at 100 mL; solution B: Tris-Gly-8M urea, with 48.048 g urea, was dissolved in solution A with the aid of an ultrasound treatment (80 kHz, 25 °C, 20–30 min); solution C: DTNB(5,5′-dithiobis-2-nitrobenzoic acid) solution (4 mg/mL), with 4 mg DTNB, was dissolved in 1 mL of solution A.

Determining the free sulfhydryl group (SH_1_) contents:

The wet samples described in Section 2.2.3 and 2.2.4 were dispersed in 5 mL of solution A, and 50 μL of solution C was added. The mixtures were kept at 25 °C for 1 h. The suspended sedimentations were removed by centrifugation (12,000× *g*) for 10 min, and the supernatants were collected to determine the absorbance at 412 nm.

Determining the total sulfhydryl group (SH_2_) contents:

The wet samples described in Section 2.2.3 and 2.2.4 were dispersed in 5 mL of solution B, and 50 μL of solution C was added. The suspended sedimentations were removed by centrifugation (12,000× *g*) for 10 min, and the supernatants were gathered to determine the absorbance at 412 nm.

The SH_1_ and SH_2_ contents of each sample were calculated as shown in Formula 1, and the disulfide bond contents were calculated as shown in Formula (2):SH1, SH2 content (μmol/g) = 73.53 × A_412_ × D/C (1)
 Disulfide bond content (μmol/g) = (SH_2_ − SH_1_)/2(2)
where A_412_ is the absorbance at 412 nm, D is the dilution factor, C is the sample concentration (mg/mL) and 73.53 is derived from 106/1.36 × 104, where 1.36 × 104 is the molar extinction coefficient.

#### 2.2.7. Molecular Weight Distribution Profiles

The molecular weight distribution of the five amylopectins was determined by high-performance size-exclusion chromatography (HPSEC) using a multi-angle laser light-scattering detector (MALLS, DAWN HELEOS II; Wyatt Technologies, Santa Barbara, USA) and a refractive-index detector (RI; Wyatt Technologies) [20]. A total of 0.25 mL of 2.0 M NaOH was added to disperse the wet amylopectin (5 mg), and then 2.0 mL of deionized water was added to completely solubilize it by oscillation at room temperature. A 0.5 µm membrane was used to filter the amylopectin solutions, and the filtrates were injected into a size-exclusion chromatography system. The high-pressure size-exclusion chromatography system was equipped with an LC-20AB pump and a manual injection valve (Hewlett-Packard, Valley Forge, PA, USA) with a 200 mL injection loop. The detection system contained a MALLS detector (Dawn EOS, Wyatt Technology, Santa Barbara, CA, USA) with a He–Ne laser source (k ¼ 658 nm), K-5 flow cell and RI detector (model 2414, Waters, Milford, MA, USA). The molecular weight distributions of the amylopectins were determined using organic SEC columns (Styrage^®®^ HMW 6E DMF 250 and 1000, 7.8 mm × 300 mm, Waters, Milford, MA, USA) at 45 °C. Dimethyl sulfoxide (DMSO) with 50 mM NaNO_3_ was used as the mobile phase at a flow rate of 0.6 mL/min. The MALLS detector was calibrated by Dextran standards (T40 and T2000). Astra software (version 5.3.4, Wyatt Technology, Santa Barbara, CA, USA) was used to handle the data in order to determine the molecular characteristics of the molecular weight.

#### 2.2.8. Chain Length Distribution Profiles

High-performance anion-exchange chromatography with pulsed amperometric detection (HPAEC-PAD) was used to determine the chain length distribution of the five amylopectins [21]. The amylopectins (100 mg) were dispersed in 50 mL of 4.0 M potassium hydroxide before the pH of the solution was adjusted to 6.0 with 6.0 M hydrochloric acid. Then, these amylopectins were debranched by the hydrolysis of pullulanase (0.1 mL, 0.5 U) at 45 °C for 24 h under constant stirring. The pullulanase was inactivated by boiling for 10 min and centrifuged at 20,000× *g*. The above-mentioned solutions were injected into the HPAEC-PAD system (Dionex Corporation, Sunnyvale, CA, USA) after being filtered through a 0.5 μm membrane filter. Data were collected and managed using Chromeleon software (version 6.50, Dionex Corporation, Sunnyvale, CA, USA).

#### 2.2.9. FTIR Spectroscopy

KBr (spectroscopic grade) was dried at 120 °C for 2h and kept in the dryer after being cooled to room temperature. Then, every sample was blended with KBr (1%, *w*/*w*) at the rate of 1:60 (*w*/*w*). The mixtures were pressed into sheets using a tablet press. Then, Fourier-transform infrared spectroscopy (Perkin–Elmer, Buckinghamshire, UK) was used to obtain the data in the transmission mode at 27 °C.

The secondary structures of glutenin in the samples, characterized by the amide I band (1600–1700 cm^−1^) in the IR spectra, were calculated according to [18]. The absorption of KBr was previously subtracted from the sample spectrum. They were assigned as follows: intermolecular β-sheets at 1612–1620 cm^−1^, β-sheets at 1625–1642 cm^−1^, α-helices at 1650–1660 cm^−1^, β-turns at 1670–1680 cm^−1^ and antiparallel β-sheets at 1680–1695 cm^−1^. The content of each secondary structure of ASG was obtained by processing Csv-format infrared data with the Peakfit software.

#### 2.2.10. ^13^C Solid-State NMR Spectroscopy

A JEOL ECZ600R 600 MHz spectrometer was used to determine the ^13^C solid-state NMR spectra at Tianjin University. The dried samples were packed into 5 mm rotors at room temperature, and the ^13^C frequency was 150.87 kHz, which corresponded to a 90° pulse width of 2.4 μs. The spinning rate of MAS was set at 15 kHz.

#### 2.2.11. X-ray Powder Diffraction (XRD) Analysis

The XRD patterns of dried samples were obtained using a D/MAX-2500 Advance diffractometer (Rigaku, Akishima-shi, Japan). The diffractometer was operated at 200 mA and 40 kV. The scanning region of the diffraction angle (2*θ*) was from 3° to 60°, and the step size was 0.02°. The counting time was 0.8 s.

#### 2.2.12. Statistical Analysis

The data presented in the paper are all expressed as the mean ± S.D. The statistical significance of differences between the control and treated samples was evaluated by two-sample t-tests of variance with Excel. Every sample was determined in triplicate. The significant differences between the means of all samples were calculated using a Dunnett’s test (*p* < 0.05).

## 3. Results and Discussion

### 3.1. The Effects of the ASG+Amylopectins on Disulfide Bond Formation

A greater content of disulfide bonds results in the more stable state of the dough’s network structure [10]. Any additional increases in the disulfide bond contents of the dough will improve the quality of the dough and corresponding food. According to [11], among wheat, corn, tapioca, sweet potato and potato starches, wheat starch–gluten dough showed the highest disulfide bonds content (3.47 μmol/g). The authors interpreted this as an indication that the small wheat starch granules could pack and build the continuous starch–gluten dough with a higher content of disulfide bonds by filling the space of the large granules. We disagree with this opinion, because there are also small granules in other starches. The formation of more disulfide bonds in gluten through the interaction of wheat amylopectin and glutenin might be the true reason.

Table 1 shows the effects of mixing five amylopectins with ASG on its disulfide bond formation before and after retrogradation. Regarding ASG, before and after the retrogradation treatment, its disulfide bond contents were 0.04 and 0.03 μmol/g, respectively, which are obviously lower than those of the original starch dough (1.76 μmol/g) [22]. The interaction of the starch and gluten might transform the secondary structure of gluten molecules into a conformation that favors the formation of disulfide bonds in the dough. The markedly low disulfide bond contents of the control groups, shown in Table 1, probably indicate that ASG might belong to the class of high-molecular-weight glutenin (HMW), which contains less cysteine [23].

Mixing the wheat, maize, cassava, mung bean and sweet potato amylopectins with ASG increased the disulfide bond contents of ASG from 0.04 to 0.31, 0.24, 0.08, 0.18 and 0.29 μmol/g, respectively. Although there was no significant difference between the mung bean and sweet potato groups, probably caused by an inhomogeneous distribution of the molecules, it is clear that mixing starches with ASG promotes the formation of disulfide bonds, especially in the case of the wheat amylopectin group. It is, moreover, interesting to note that the retrogradation treatment, as shown in Table 1, increased the disulfide bond content of the wheat amylopectin group from 0.31 to 0.55 μmol/g and decreased that of the maize amylopectin group from 0.24 to 0.16 μmol/g. It is well-known that a large number of hydrogen bonds are formed between starch molecules during retrogradation. Thus, the speculation that hydrogen bond formation promotes or restrains disulfide bond formation is debatable [3,24]. Hydrogen bonds only favored the disulfide bond formation of ASG in the retrogradation treatment of the wheat/cassava amylopectins, and in the other three kinds of amylopectin samples, the contents of disulfide bonds were obviously reduced by cold storage. The key to this should be the secondary structure of amylopectins. The special characteristic of wheat amylopectin was further investigated according to the results of the molecular weight and distribution, IR, ^13^C solid-state NMR and X-ray diffraction analyses.

The lower disulfide bond content of the control sample, after being heated by autoclaving and cooled by refrigeration, as shown in Table 1, demonstrates that heating above 90 °C promotes the reduction reaction of the disulfide bonds formed at 35–90 °C in ASG [25].

### 3.2. The Average Molecular Weight (M_w_) and Chain Length Distribution

According to [11,26,27,28,29], the *M_w_* values of wheat, maize, cassava, mung bean and sweet potato amylopectins are 0.29~3.49, 0.70~0.98, 2.42~2.74, 0.30~0.69 and 0.69~1.65 × 10^8^ g/mol, respectively. The *M_w_* of wheat amylopectin is the highest, and that of mung bean amylopectin the lowest. Table 2 shows the *M_w_* values of wheat, maize, cassava, mung bean and sweet potato amylopectins in our experiments, which are 0.035, 0.022, 0.038, 0.006 and 0.020 × 10^8^ g/mol, respectively. The lower *M_w_* values in our paper can be attributed to the molecular cleavage of the amylopectins during the freeze–thaw process, which can also be verified by the appearance of a large number of small molecules in all the samples shown in Table 2. Water washing does not remove these small molecules, probably because of a type of high-affinity binding between them and certain macromolecules. Cassava amylopectin has the largest *M_w_* value, while mung bean amylopectin has the lowest one. There is no correlation between the *M_w_* values and disulfide bond contents. Again, no correlation between the dispersity values and disulfide bond contents is shown in Table 1. The dispersity of the mung bean and sweet potato amylopectins reaches 34.97 and 32.91, respectively, indicating that they are less resistant to freezing and thawing treatment.

Table 3 shows the chain length distribution of the amylopectins from the different cultivars. Only the wheat and mung bean amylopectins have chains containing more than 15 glucose residues, and the appearance of these residues has no relationship with the disulfide bond contents. Upon closer inspection, it can be found that wheat amylopectin has the highest proportion of 9–12 residues in the glucose chain, indicating that the secondary structure of these chains might favor the formation of disulfide bonds. This will be discussed further in Section 3.3, Section 3.4 and Section 3.5 below.

### 3.3. Fourier Transform Infrared Spectroscopy Analysis

Figure 1 shows the FT-IR spectra of the different amylopectins mixed with and without ASG. The band at 3301 cm^−1^ is assigned to the N-H stretching vibration of ASG, and the two bands at 3285 (for maize) and 3298 cm^−1^ (for wheat, cassava, mung bean and sweet potato) in Figure 1 are due to the O-H stretching vibration of the amylopectins [30,31,32]. The clearly lower wavenumber shown in Figure 1b for maize amylopectin in this field might be attributed to the fact that it has the most chains of single glucose residues, as shown in Table 3. Hydrogen bonds are more likely to form between these chains, causing the wavenumber of the maize amylopectin shift to the low-frequency region [33]. It has been noted that the mixing of starches and gluten plays an important role in the formation of gluten disulfide bonds [11], and our results in Table 1 suggest that, probably, only the amylopectins serve a vital function. Ogawa et al. (1998) [32] believe that the smooth degree of the region is related to the retrogradation degree of starch. Thus, the effects of amylopectin retrogradation on disulfide bond formation could be deduced by comparing the intensities of the different amylopectins in this band. The red, blue and purple lines in Figure 1 represent amylopectins, amylopectins + glutenin complex and retrograded amylopectins + glutenin, respectively. In all the samples in Figure 1 apart from the cassava group, the smoother shape of the blue lines compared to the red ones indicate that the mixing of amylopectins and ASG induces the formation of hydrogen bonds among the amylopectin chains or between the amylopectin and glutenin molecules. The higher disulfide bond contents of these samples suggest that these hydrogen bonds favor disulfide bond formation in ASG. These hydrogen bonds might accelerate the rearrangement of the polymer network, thus leading to the disulfide exchange reaction, increasing the content of disulfide bonds [34]. However, there is no relationship between the disulfide bond contents and the change in the smooth degree of the infrared absorption peak at ~3301 cm^−1^ in the field for any of the samples. The smooth degrees of the infrared absorption peak at ~3301 cm^−1^ are reduced for the wheat, maize and sweet potato groups, increased for the cassava group and unchanged for the mung bean group.

The typical protein bands for amide I (80% C=O stretch, 10% C-N stretch) at ~1633 cm^−1^ and amide II (60% N-H bend, 30% C-N stretch and 10% C-C stretch) at ~1529 cm^−1^ are all very weak, and bands only appear at around ~1644 cm^−1^. Such bands are assigned to the H-O-H bending mode of water [31], because for all the amylopectins and complex samples, they are nearly same. The weaker intensity of the band for wheat amylopectin + ASG demonstrates that water molecules are involved in the formation of disulfide bonds during mixing, so that the weaker the band is, the higher the disulfide bond content will be, and vice versa (marked by a red dotted arrow in Figure 1). We can compare this with [35]. These authors’ bands for amide I and amide II of glutenin were clear, implying that ASG had a higher degree of aggregation. During retrogradation, the intensity degrees of the bands had no relationship with the disulfide bond contents of ASG.

The secondary structure of protein can be determined by the analysis of the Fourier-deconvoluted data. Five bands at approximately 1605, 1632, 1652, 1680 and 1695 cm^−1^ are characteristic of aggregations with the β-sheet, extended β-sheet, α-helical, β-turn and extended β-sheet conformation structures, respectively [36]. The changes in the secondary structure of ASG mixed with and without different amylopectins before and after retrogradation are shown in Table 4. The secondary structure of gluten in the non-hydrated state is estimated to be 17% α-helix, 39% β-sheet, 14% β-turn and 30% random [37]. ASG contains 0% α-helix, 50.62% intermolecular β-sheet, 47.14% intra-molecular aggregation extended β-sheet, 0% β-turn and 2.24% random, a composition which is obviously different from that of gluten. Through mixing with amylopectins, the α-helix, β-turn and intra-molecular aggregation extended β-sheet contents of ASG increase, and the intra-molecular aggregation extended β-sheet and random coil contents decrease. During retrogradation, the reduction in the α-helix contents could be regarded as an indicator of disulfide bond content increase, and the decrease in the intra-molecular aggregation extended β-sheet, coinciding with the increase in the random coil contents, points to a lower disulfide bond content.

We therefore focused our attention on the symmetrical stretching vibration of the disulfide bonds at 500–510 (assigned to gauche-gauche-gauche conformation), 515–525 (gauche-gauche-trans conformation) and 535–545 cm^−1^ (trans-gauche-trans conformation), respectively [38]. There are no such bands in Figure 1, implying that disulfide bonds are vibrationally bound or are enwrapped in the secondary structure of ASG.

The characteristic band at ~432 cm^−1^ (marked with a dashed black arrow) in Figure 1b, d and e, being assigned to C-C-O deformation vibrations [39], is absent in the wheat and cassava groups. It probably is associated with molecules with a high molecular weight, because the molecular weights of these two samples are obviously higher than those of the other samples in Table 3. As ASG is combined with amylopectin with a higher molecular weight, the spatial structure of the ASG molecules may be changed and stabilized by the water molecules, so that the structure is conducive to the formation of disulfide bonds. There is no useful information regarding the other bands.

### 3.4. ^13^C Solid-State NMR Spectra of Different Amylopectins Mixed with and without ASG

The original intention of this paper was to explore the mechanism by which wheat amylopectin promotes the formation of disulfide bonds in gluten. The chemical bonds newly formed and certain secondary structure of ASG to enhance disulfide bonds would be identified by comparing the results of the ^13^C solid-state NMR spectra of ASG mixed with and without different amylo-pectins, and those of mixed samples before and after retrogradation. Figure 2 shows the ^13^C solid-state NMR spectra of the different amylopectins mixed with and without ASG. According to [40], resonances at 0–40 ppm are assigned to alkyl groups in the protein side chains and lipids, with those at 40–65 ppm assigned to alkyl groups in the main protein chains, those at 65–105 ppm assigned to starch, those 125–135 ppm assigned to protein aromatic moieties and lipid olefinic carbons, and those at 165–185 ppm assigned to carbonyl groups in the proteins and lipids. The absence of intense aliphatic resonance at 29 ppm for the lipid in Figure 2 indicates that there is no lipid in any of the samples [41].

Table 5 shows the detailed information about the resonance changes of all the samples. Unusually, the typical resonances for glutenin at ~177 ppm (Q_δ_), 172 ppm (backbone C=O), 60 ppm (P_α_, T_α_, S_β_), 52 ppm (Q_α_, L_α_, A_α_), 48 ppm (P_δ_), 42 ppm (G_α_, L_β_), 30 ppm (Q_γ_, P_β_) and 25 ppm (P_γ_, L_γ_) and the shoulders at 19–21 ppm (methyls L_δ_, T_γ_, A_β_) are all very weak or absent with respect to ASG, as shown in Figure 2 [42,43,44]. However, the intensities of these resonances for oligosaccharides or starches are obviously strong, suggesting that ASG is a kind of glycoprotein instead of a glutenin+starch complex, because these starches in gluten were hydrolyzed during the extraction of glutenin in 0.1% NaOH and removed by dialysis. In order to assign the resonances of ASG accurately, the weak peaks are enlarged in Figure 2 and assigned in Table 5. Based on [42,43,44,45,46], the resonances of ASG are assigned as follows: 174.1 ppm (Q_δ_ linked with N-glycosidic bond), 172.2 ppm (backbone C=O), 132.6 ppm (Y_δ_), 131.4 ppm (Y_γ_), 128.5 ppm (Y_ε_), 103.4 + 95.1 and 82.1 ppm (C1 and C4 of the oligosaccharides in the amorphous region), as well as 31.9 ppm (Gln C_β_). The resonances for C2, 3, 5, 6 of the oligosaccharides in all the samples, as shown in Table 5, remain unchanged, indicating that no new chemical bonds are formed on these carbon atoms during mixing and retrogradation. For the wheat amylopectin in Table 5, the weak resonances at 173.8 ppm (Q_δ_ linked with an N-glycosidic bond without Tyr), 171.5 ppm (hydrogen-bonded backbone C=O), 32.6 ppm (Gln C_γ_) and 31.5 ppm (hydroxyproline (HYP) C_β_) show that, compared to ASG, wheat amylopectin combines with a kind of protein containing no Tyr and having a HYP in the side chains. When ASG is mixed with wheat amylopectin, as shown in Figure 2a (blue line) and Table 5, the formation of disulfide bonds is enhanced. During the mixing process, the resonances for Q_δ_ and C=O shift to the lower and higher fields, respectively, and those of Y_γ_ (131.4 ppm) in ASG and HYP C_β_ (31.5 ppm) in the wheat amylopectin disappear. The former may be caused by the formation of hydrogen bonds, and the latter may form a covalent bond between the two amino acids. Dough formation is a hydration process, and hydrogen bonds should form between the Q_δ_/backbone C=O and water. According to the results in Table 4, the α-helix, intra-molecular aggregation extended β-sheet and β-turn contents of ASG are increased at the expense of the intermolecular β-sheet and random coils contents. Thus, it is suggested that the covalent bonds between the Y_ζ_ of ASG and HYP C_γ_ in wheat amylopectin promote α-helix formation. The Y_γ_ of ASG and HYP C_β_ are buried in the α-helix structures, which leads to the loss of their resonances. This is an important step through which wheat amylopectin promotes the disulfide bond formation of ASG. The resonances at 161.7/160.3/159.6/158.8ppm are assigned to Tyr C_ζ_ (Y_ζ_) [47], and their weak intensities for the wheat amylopectin+glutenin group indicate that most of them were buried in the α-helix structures during dough formation. As the dough of ASG and wheat amylopectin was subjected to retrogradation for 7d at 4 °C, as shown in Figure 2a (purple line) and Table 5, Q_δ_ linked with the N-glycosidic bond shifts to a higher field, indicating that the hydrogen bonds between Q_δ_ and water disappeared during retrogradation. This can also be proved by the precipitation of the water on the surface of the sample after retrogradation. The reappearance of resonances at 131.4 ppm (Y_γ_) and loss of resonances at 132.5 ppm (Y_δ_) suggest that they probably combine with sulfhydryl and are involved in disulfide bond formation. The resonance for the hydrogen-bonded backbone C=O at 171.3 ppm is always present, showing that the hydrogen bonds between C=O and water remain unchanged. It is noteworthy that the increase in the resonances for Y_ζ_ at 159.6/158.8 ppm appears in the retrograded wheat amylopectin + ASG group in Figure 2f, implying that these Y_ζ_ that were originally involved in the formation of the covalent bonds return to their original hydroxyl state. The cleavage of the covalent bonds in the dough is probably caused by the autoclaving treatment before retrogradation. Compared with wheat amylopectin, maize amylopectin also combines with protein containing Gln, but no resonance at 31.5 ppm for HYP C_γ_ appears in Figure 2b and Table 5. When it is mixed with ASG, the difference in the resonance is present at 131.7 ppm for Y_γ_, but there is an absence of resonance at 132.6 ppm for Y_δ_, indicating that -S of the sulfhydryl of cysteine might combine with Y_δ_, unlike that of wheat amylopectin, Y_γ_. The change degree of the secondary structure of ASG in the maize amylopectin + ASG group is lower than that of the wheat amylopectin group, except for the β-turn contents. During the retrogradation of the maize amylopectin + ASG, the procedure leads to reductions in the disulfide bond contents from 0.24% to 0.16%, as shown in Table 1 and Table 2. The secondary structures of the α-helix and intra-molecular aggregation extended β-sheet contents increase, and those of the intermolecular β-sheet and β-turn contents decrease, clearly differing from those of wheat amylopectin group. It is worth noting that there is no trace of protein in the cassava amylopectin in Figure 2c (marked with dashed arrows) and Table 5. This may be the fundamental reason explaining why it cannot promote the formation of a large number of glutenin disulfide bonds during dough formation. When cassava amylopectin is mixed with ASG, as shown in Figure 2c and Table 5, all the resonances for Tyr at 132.6/131.4/128.5 ppm disappear. They are probably buried in the new secondary structures of intra-molecular aggregation extended β-sheets and β-turns. After retrogradation, the most disulfide bonds form in the cassava amylopectin group, and the single resonance at 131.7/103.4 ppm indicates that the Tyr of glutenin and C1 of cassava amylopectin are involved in the crystal formation of retrograded cassava amylopectin + ASG. The absence of the resonances at 161.7/160.3/159.6/158.8ppm for Y_ζ_ in this group, as shown in Figure 2f, suggests that the hydroxyls of the Tyr of ASG are involved in the formation of hydrogen bonds among the complexes. For the mung bean and sweet potato amylopectin groups, the increase and decrease in the disulfide bond contents occur during the mixing of the amylopectins and ASG and their retrogradation, as shown, respectively, in Table 1 and Table 2. The common features of the mung bean and sweet potato amylopectins in Figure 2d,e and Table 5 include the higher resonances for the C1 of amylopectin and a lack of resonance for the alkyl groups in the protein side chains. The substance combined with those amylopectins might be short peptides rather than proteins. When the two amylopectins are mixed with ASG, the appearance of resonances at ~175/173/171 ppm in Figure 2d,e and Table 5 suggests that glutathione disulfide (GSSG) is generated [48]. After retrogradation, the resonances for this GSSG disappear, indicating that it depolymerizes in the process.

Whether in mixture or in retrogradation, the absence of resonances for Tyr shows that they are all buried in the secondary structures of the complexes. The enhancement of the resonances at 160.3 and 161.7 ppm for the retrograded mung bean amylopectin + ASG and sweet potato amylopectin + ASG groups, respectively, shows that the hydroxyls of the Y_ζ_ of ASG are converted to free ones. The sharply reduced content of intra-molecular aggregation extended β-sheet secondary structure for mung bean amylopectin + ASG from 60.47% to 48.32% before and after retrogradation implies that the formation of covalent bonds between tyrosines is a prerequisite for the formation of intramolecular disulfide bonds in ASG, which agrees well with the findings of [49]. The findings for mung bean and sweet potato amylopectin and ASG complexes before and after retrogradation shed light on the nature of the effects of the addition of mung bean and sweet potato starches on the dough properties. Increased glutathione dimer reductase activity caused by the increase in the helix structures of ASG, as shown in Table 4, might have occurred in the mung bean/sweet potato amylopectin + ASG groups during refrigeration. Dimer glutathione reductase, showing an inhomogeneous sample distribution in the mung bean and sweet potato amylopectin groups, might lead to the low reproducibility of the disulfide bond content of the same sample, resulting in no significant difference compared with the blank in Table 1. Dough containing these two amylopectins should not be stored by refrigeration.

### 3.5. X-ray Diffraction of the Different Amylopectins Mixed with ASG

It is well-known that the crystal patterns of different granules can be classified into A-, B- and C-types using XRD spectra, and the typical patterns have their own characteristic diffraction angles, with the A-type 2*θ* at ~15^o^ (strong), ~17° (unresolved), ~18° (unresolved) and ~23° (strong); the B-type 2*θ* at ~5.6°, ~15° (small), ~17^o^ (strong), ~20° (small), ~22° (small) and ~24° (small); and the C-type 2*θ* at ~17° (strong), ~23° (strong)~, 5.6° (small) and 15° (small) [46]. A- and B-type starches are mostly derived from cereal crop seeds and some plant tubers, respectively, and C-type starch exists in some legume seeds and some plant rhizomes [50]. Figure 3 shows the X-ray diffractions of the different amylopectins mixed with ASG before and after retrogradation. The diffraction angles of ASG are at 2*θ* 13.32°, 15.58°, 17.74°, 18.16°, 20.16° and 22.96°, and those of the wheat amylopectin are located at 2*θ* 15.76°, 17.46° and 20.16° in Figure 3a. When they are mixed, only diffraction angles at 2*θ* 17.38° and 20.10° remain, and retrogradation has little effect on them. For the maize amylopectin in Figure 3b, different angles appear at 2*θ* 15.62°, 17.50° and 20.28°, and the same results as those of wheat amylopectin upon mixture and retrogradation are obtained. The cassava amylopectin in Figure 3c, the only amylopectin not combined with protein, shows the typical diffraction angles at 2*θ* 17.44° and 23.00°, which are same as those of the sweet potato amylopectin purified by the hydrolysis of proteases and lipases [51]. Thus, it is debatable as to whether we should classify all amylopectin crystalline structures as A-type starch [52]. The mixture and retrogradation of cassava amylopectin + ASG resulted in the maintenance of the crystal type of ASG, with a lower intensity of the angle at 2*θ* 20.12°. For the mung bean amylopectin in Figure 3d, its typical diffraction angles are located at 2*θ* 17.14° and 22.42°. After being mixed and retrograded with ASG, the diffraction angles for the complexes converted to 2*θ*~17.14°, ~19.86° and ~22.34°. Compared with the diffraction angles of ASG, the most obvious change is the absence of angles at 2*θ* 13.32° and 15.58°. For the sweet potato amylopectin in Figure 3e, its typical diffraction angles are nearly the same as those of ASG in Figure 3a. Its mixture and retrogradation do not change the crystal type of the complexes, as mixing induces the higher intensity of the angle at 2*θ*~15°, and retrogradation causes it to disappear. If only the groups with significant differences are considered, there is a positive/negative correlation between the peak intensity at the diffraction angle of 2*θ*~20°/23° in Figure 3 and the disulfide bond contents in Table 1 and Table 2. The findings of this paper lead us to question the opinion that starch with amylopectin possessing more short-branch chains is the A-type and that the one with more long-branch chains is the B-type [53], as they are just the diffraction of the starch–protein complexes.

### 3.6. The Possible Mechanism of Disulfide Bond Formation When ASG Is Mixed and Co-Retrograded with Different Amylopectins

After the comprehensive analysis of the changes in disulfide bond formation, the distribution characteristics of the starch molecular weight and chain length and the changes in the ^13^C solid-state NMR and IR results during the mixing and retrogradation of the different amylopectins and ASG, we speculated on the possible mechanism of the influence of wheat amylopectin on the disulfide bond formation of ASG, as shown in Figure 4. Wheat and cassava amylopectin are the two of the five amylopectins that promote disulfide bond formation in ASGs during both mixing and retrogradation. However, wheat amylopectin promotes the formation of more disulfide bonds in ASG. It is different from the other four kinds of amylopectin in the features of the molecular weight, being greater than 1.3 million Da, and the side chains, with chain lengths of 9–12 glucose residues, accounting for the highest proportion and presence of hydroxyproline. When ASG is mixed with wheat amylopectin, the side chains interact with glutenin by way of dehydration condensation between the C_ζ_ of the Tyr in glutenin and C_γ_ of the Hyp in wheat amylopectin, as shown in Figure 4a, which promotes the appearance of α-helix in ASG, as shown in Figure 4b. This speculation is consistent with the latest literature results [54], suggesting that the proportion of α-helices, β-turns and antiparallel β-sheets increases when glutenin is mixed with starch at low temperatures. At the same time, intra-molecular aggregation extended β-sheet and β-turn contents increase at the expense of the intermolecular β-sheet content. At this stage, sulfhydryls dissociated from the Cys of ASG combine with Tyr at C_γ_-C_δ_, and this binding may be accomplished with the help of a certain thioltransferase. Based on this, disulfide bonds are formed at this binding site under the combined action of the mixing forces and the movement of the water molecules. These disulfide bonds should be formed in the structure of intra-molecular aggregation extended β-sheets and β-turns at this stage, because the increase in their contents coincides with the increase in the contents of the disulfide bonds. Additionally, we can thus speculate that, with the help of water movement, the newly formed α-helix structure of ASG originates from a random coil structure, which is pulled by the helix of the wheat amylopectin attached to ASG in a spiral movement, as shown in Figure 4b. According to the results in Table 5, the hydrogen bonds between the C6 of wheat amylopectin and C=O of glutamine in ASG might stabilize the helical structure, providing it with more potential to produce intra-molecular aggregation extended β-sheet and β-turn structures. When the complex of wheat amylopectin and ASG is retrograded in a refrigerated environment, as shown in Figure 2 and Figure 4c, the Tyr buried in the α-helix structure of ASG reappears as the partial α-helix structure transforms into intermolecular β-sheet and β-turn structures. The source of this transformation is the formation of hydrogen bonds between the wheat amylopectin molecules during retrogradation, resulting in a reduction in the original helical structure of the wheat amylopectin attached to ASG. Then, hydrogen bonds form between the C_ζ_ of Tyr in glutenin and C_γ_ of Hyp in wheat amylopectin, as shown in Figure 4c, which produces the environmental conditions required for disulfide bond formation. Different from those formed in the first stage, these disulfide bonds are bonded to the C_δ_ of Tyr in glutenin. This difference is highly worthy of in-depth study. The increase in the intermolecular β-sheet and β-turn structure of ASG at the retrogradation stage is perfectly understandable, because the hydrogen bonds between the glutenin and wheat amylopectin molecules replace those between the glutenin/amylopectin and water. The straightening of these protein molecules provides the cysteine with a greater opportunity to oxidize and form disulfide bonds. The main limiting factors that determine the content of disulfide bonds include the activities of sulfhydryl transferase and sulfhydryl oxidase, the spatial distance between intramolecular or intermolecular cysteines, environmental pH, etc. High-molecular-weight amylopectin can promote and stabilize the formation of disulfide bond during retrogradation, while low-molecular-weight amylopectin may activate disulfide bond reductase in the stage, resulting in a sharp decrease in the disulfide bond contents, as shown in Table 2 and Table 3.

For the mung bean and sweet potato amylopectins, being two commonly used starches, a surprising discovery is that their addition promotes the generation of glutathione dimers in ASG during the mixing treatment. However, these dimers depolymerize during starch retrogradation, as shown in Table 2. Transitions between glutathione dimers and monomers may occur, as described in the literature [55]. This finding explains the sharp decline in the quality of dough with the addition of green bean flour or sweet potato flour after cold storage. Whether or not wheat protein disulfide isomerase (wPDI) is involved in this process needs to be studied further. In actual application, wheat starches with high contents of hydroxyproline should be selected so as to obtain elastic dough. The results of this paper offer a new way to screen wheat varieties that could produce more elastic dough.

## 4. Conclusions

Wheat amylopectin is the starch that promotes the most significant disulfide bond formation of ASG among the five amylopectins, and it is characterized by the presence of hydroxyproline, a molecular weight greater than 1.3 million Da, and the highest proportion of side chains with chain lengths of 9–12 glucose residues. The disulfide bonds formed by the interaction of ASG and mung bean or sweet potato amylopectin due to glutathione polymer linked by disulfide bonds in the mixing process will be partially reduced during cold storage. The characteristic crystal types of different starches are correlated with the amylopectin + glutenin complexes.

## Figures and Tables

**Figure 1 foods-12-00414-f001:**
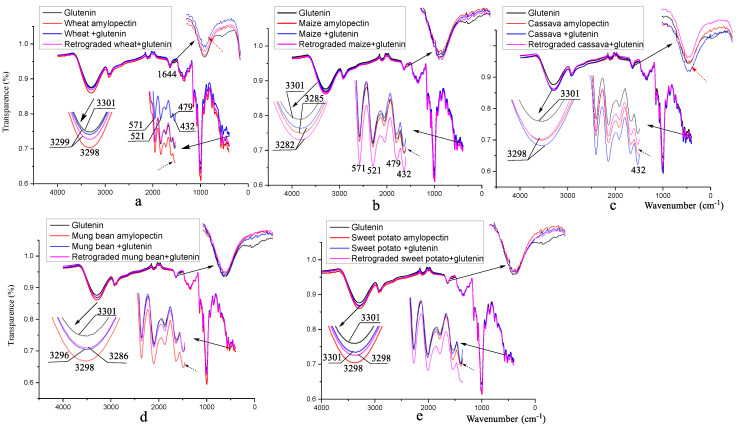
FT-IR spectra of different amylopectins mixed with and without alkali-soluble glutenin. The subfigures are the graphs of a particular peak magnified by a factor of 5–10. (**a**) wheat amylopectin group, (**b**) maize amylopectin group, (**c**) cassava amylopectin group, (**d**) mung bean amylopectin group, (**e**) sweet potato amylopectin group. Result for glutenin (alkali-soluble glutenin) in Figure 1a–e is from the same sample, addition of it in every figure is to facilitate comparative analysis.

**Figure 2 foods-12-00414-f002:**
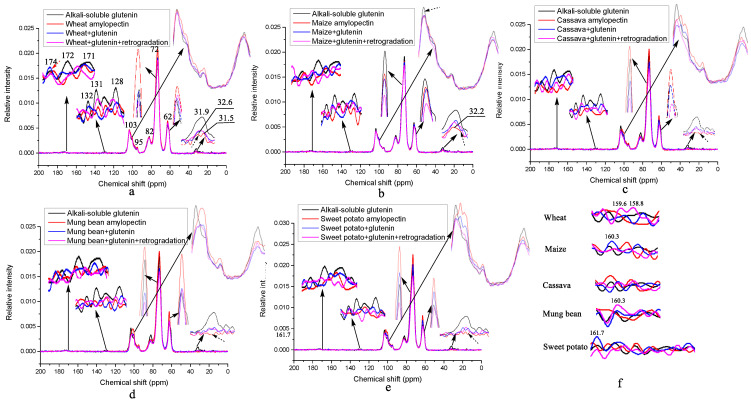
^13^C solid-state NMR spectra of different amylopectins mixed with and without alkali-soluble glutenin. The subfigures are the graphs of a particular peak magnified by a factor of 5–10. (**a**) wheat amylopectin group, (**b**) maize amylopectin group, (**c**) cassava amylopectin group, (**d**) mung bean amylopectin group, (**e**) sweet potato amylopectin group, (**f**) the changes of resonance for Y_ζ_ (Tyr) of alkali-soluble glutenin during mixing and retrogradation. Result for alkali-soluble glutenin in Figure 2a–f is from the same sample, addition of it in every figure is to facilitate comparative analysis.

**Figure 3 foods-12-00414-f003:**
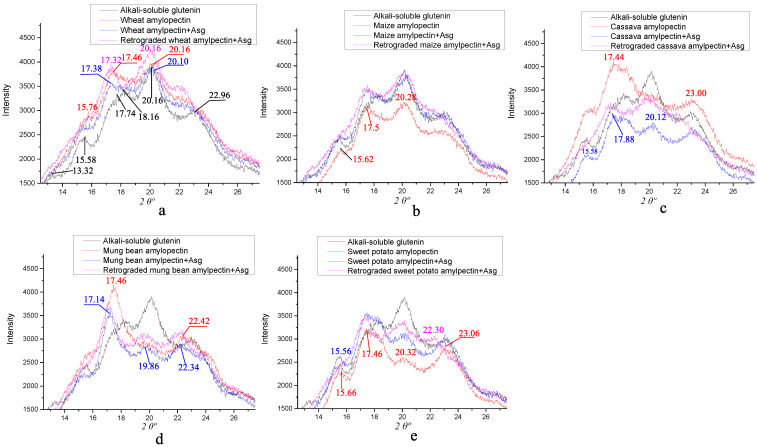
X-ray diffraction of different amylopectins mixed with alkali-soluble glutenin before and after retrogradation. (**a**) wheat amylopectin group, (**b**) maize amylopectin group, (**c**) cassava amylopectin group, (**d**) mung bean amylopectin group, (**e**) sweet potato amylopectin group. Result for alkali-soluble glutenin in Figure 3a–e is from the same sample, addition of it in every figure is to facilitate comparative analysis.

**Figure 4 foods-12-00414-f004:**
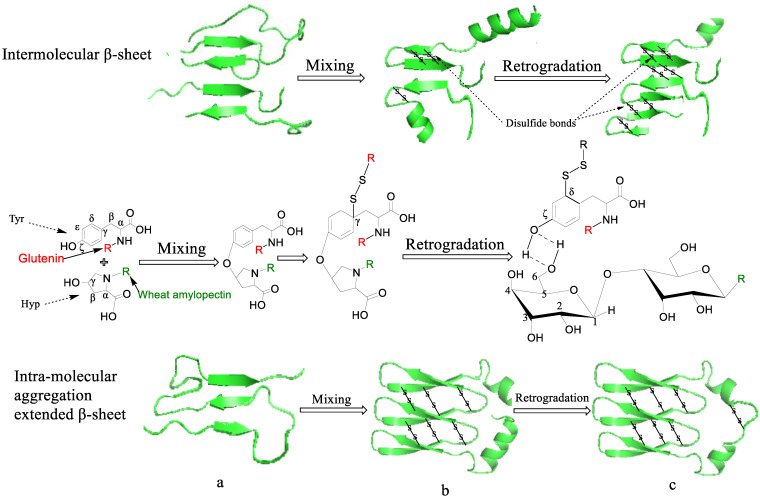
Proposed schematic diagram of disulfide bond formation during the mixing and retrogradation of wheat amylopectin + alkali-soluble glutenin. (**a**). initially secondary structure of alkali-soluble glutenin, (**b**) secondary structure of alkali-soluble glutenin being mixed with wheat amylopectin, (**c**) secondary structure of alkali-soluble glutenin mixed with wheat amylopectin after retrogradation.

**Table 1 foods-12-00414-t001:** Disulfide bond contents of alkali-soluble glutenin mixed with five amylopectins before and after retrogradation (μmol/g).

Samples	Control	Wheat	Maize	Cassava	Mung Bean	Sweet Potato
Before retrogradation	0.04 ± 0.00 a	0.31 ± 0.02 b,**	0.24 ± 0.02 c,**	0.08 ± 0.00 d,*	0.18 ± 0.03 e	0.29 ± 0.11 b
After retrogradation	0.03 ± 0.00 a	0.55 ± 0.03 b,**	0.16 ± 0.03 c,*	0.26 ± 0.05 d	0.07 ± 0.01 e	0.19 ± 0.04 f

* *p* ˂ 0.05, ** *p* ˂ 0.01. The objects of significant comparison for the samples before and after retrogradation are the corresponding control groups. Values of all the samples followed by the same or a different letter are not significantly different or significantly different at the 5% level of significance.

**Table 2 foods-12-00414-t002:** Molecular weight characteristics of different amylopectins isolated by freeze–thawing + 0.5% NaCl dissolution methods.

Molecular Characteristics	Amylopectins				
	Wheat	Maize	Cassava	Mung Bean	Sweet Potato
Retention time (min)	12.84 (51%) 16.52 (40%) 19.45 (9%)	13.34 (47%) 17.05 (27%) 19.45 (26%)	12.75 (39%) 16.50 (35%) 19.50 (26%)	16.90 (81%) 19.50 (19%)	12.96 (82%) 19.64 (18%)
Mn (g/mol)	1387006 18672 828	326426 12381 942	1501812 16784 682	19894 611	62126 713
Mw (g/mol)	3503982 63486 1041	1211463 22550 1314	3881679 61698 910	695605 854	2044753 1112
Mp (g/mol)	3509755 38910 1085	1337876 21017 1411	3938537 40091 1055	24461 1014	3047294 853
Polydispersity	2.53 3.40 1.26	3.71 1.82 1.39	2.58 3.68 1.33	34.97 1.40	32.91 1.56

**Table 3 foods-12-00414-t003:** Chain length characteristics of different amylopectin isolated by freeze–thawing + 0.5% NaCl dissolution methods.

Chain Length (Glucose Number)	Amylopectins				
	Wheat	Maize	Cassava	Mung Bean	Sweet Potato
1	3.6	10.89	3.55	5.57	4.41
2	20.04	9.48	18.66	30.56	18.07
3	6.96	14.07	11.78	5.56	12.93
4	2.08	16.68	15.01	4.53	3.13
5	11.09	14.06	10.47	7.33	14.78
6	8.34	15.13	15.89	4.84	12.64
7	10.11	9.96	11.68	4.68	14.92
8	8.15	5.11	6.75	4.28	10.32
9	7.19	2.47	2.93	4.07	5.16
10	5.57	1.3	1.57	3.77	2.01
11	4.35	0.51	0.85	3.54	1.06
12	3.33	0.25	0.42	3.19	0.41
13	2.73	0.11	0.28	2.86	0.17
14	1.91		0.16	2.65	
15	1.39			2.17	
16	1.03			1.91	
17	0.74			1.63	
18	0.55			1.39	
19	0.39			1.17	
20	0.28			0.98	
21	0.18			0.82	
22				0.69	
23				0.56	
24				0.45	
25				0.35	
26				0.26	
27				0.19	

**Table 4 foods-12-00414-t004:** The secondary structures of alkali-soluble glutenin mixed with and without different amylopectins before and after retrogradation.

Samples	α-Helix Content (%)	Intermolecular β-Sheet Content (%)	Intra-Molecular Aggregation Extended β-Sheet Content (%)	β-Turn Content (%)	Random Coils Content (%)
ASG (Asg)	0.00	50.62	47.14	0.00	2.24
Wheat amylopectin + Asg	1.17	36.71	59.82	2.31	0.00
Retrograded wheat amylopectin + Asg	0.15	37.48	59.39	2.97	0.00
Maize amylopectin + Asg	0.74	42.87	52.02	4.38	0.00
Retrograded maize amylopectin + Asg	0.93	36.89	59.53	2.65	0.00
Cassava amylopectin + Asg	0.00	38.73	59.46	1.81	0.00
Retrograded cassava amylopectin + Asg	0.00	37.90	60.25	1.86	0.00
Mung bean amylopectin + Asg	0.53	36.91	60.47	2.09	0.00
Retrograded mung bean amylopectin + Asg	0.76	43.50	48.32	2.80	4.62
Sweet potato amylopectin + Asg	0.00	40.70	55.71	3.59	0.00
Retrograded sweet potato amylopectin + Asg	0.11	38.46	58.77	2.66	0.00

**Table 5 foods-12-00414-t005:** Assignments of the high-resolution ^13^C solution-state NMR spectra of ASG before and after being mixed with different amylopectins, both before and after retrogradation.

Samples	Chemical Shift and Assignments (ppm)						
	Carbonyl Groups	Protein Aromatic Moieties	C1 of Oligosaccharide or Starch	C4 of Oligosaccharide or Starch	C2, 3, 5 of Oligosaccharide or Starch	C6 of Oligosaccharide or Starch	Alkyl Groups in Protein Side Chains
ASG	174.1, 172.2	132.6, 131.4, 128.5	103.4, 95.1	82.1	73.0	62.7	31.9
Wheat amylopectin	173.8, 171.5	nd	103.4, 94.8	82.0	73.2	62.5	32.6, 31.5
Wheat + glutenin	174.6, 171.3	132.5, 128.3	103.4, 95.1	82.2	73.1	62.6	32.0
Wheat + glutenin + retrogradation	173.2, 171.3	131.2, 128.9	103.3, 95.1	82.5	73.1	62.4	32.3
Maize amylopectin	173.3, 171.3	nd	103.3, 95.0	82.3	73.0	62.7	32.2
Maize +glutenin	173.3, 171.2	131.7, 129.0	103.2, 94.9	82.4	73.1	62.5	31.9
Maize + glutenin + retrogradation	173.6, 171.9	130.8, 129.6	103.4, 95.0	82.7	73.1	62.6	32.1
Cassava amylopectin	Ignorable	nd	103.2, 95.1	82.7	73.1	62.6	nd
Cassava +glutenin	173.0, 171.0	nd	103.3, 94.8	82.4	73.0	62.7	32.1
Cassava + glutenin + retrogradation	173.8, 172.2	131.7	103.4	82.0	73.2	62.4	31.8
Mung bean amylopectin	174.3, 171.6	nd	101.3, 95.3	82.0	73.1	62.6	nd
Mung bean + glutenin	175.3, 173.9, 171.3	nd	100.9, 95.0	82.3	73.0	62.6	nd
Mung bean + glutenin + retrogradation	171.8	nd	101.9, 94.9	82.3	73.1	62.4	nd
Sweet potato amylopectin	172.7	nd	102.0	82.3	73.0	62.7	nd
Sweet potato + glutenin	175.4, 173.4 172.3	nd	103.4, 94.9	82.4	73.0	62.7	nd
Sweet potato + glutenin + retrogradation	Ignorable	nd	103.3	82.1	73.2	62.5	nd

nd: not detectable.

## Data Availability

Original data can be obtained from the corresponding author.

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
