# Peer review of "Effects of Amylopectins from Five Different Sources on Disulfide Bond Formation in Alkali-Soluble Glutenin"

_foods, 2023, doi:10.3390/foods12020414_

Round 1
Reviewer 1 Report
I have reviewed the submitted manuscript (foods-2112508; Effects of amylopectins from five different sources on disulfide bond formation in alkali-soluble glutenin). Paper is well structured and all experimental procedures are clearly described. Manuscript contains some interesting observations, however, there are several additional comments regarding this manuscript:
Line 4: Authors Name Part → use * instead of † for showing Corresponding author.
Abstract: abstract should state briefly the purpose of the study undertaken, brief mention of experimental aspects (without using abbreviations), highlights of the results and important conclusions based on the obtained results. Therefore, it is suggested that the Abstract should be improved with more data.
Keywords→ arrange in alphabetical order.
Statistical analysis: You didn’t compare the significant difference between treatment means by any method such as Duncan’s test or Tukey? Why?
Table 1 and Table 2: Please compare the significant difference between treatment means and show difference with letters.
Table 4 → use journal format (Remove left, right and inside borders)
Conclusions: this section given here are do not reflect what had been achieved including many speculations. Hence these need to be suitably modified.
Author Response
Reviewers' comments:Reviewer #1: I have reviewed the submitted manuscript (foods-2112508; Effects of amylopectins from five different sources on disulfide bond formation in alkali-soluble glutenin). Paper is well structured and all experimental procedures are clearly described. Manuscript contains some interesting observations, however, there are several additional comments regarding this manuscript:
(1) Line 4: Authors Name Part → use * instead of † for showing Corresponding author.We must apologize for our carelessness about this point.
We have replace “†” by “*” at the position.
(2) Abstract: abstract should state briefly the purpose of the study undertaken, brief mention of experimental aspects (without using abbreviations), highlights of the results and important conclusions based on the obtained results. Therefore, it is suggested that the Abstract should be improved with more data.
We have added more data in abstract.
(3) Keywords→ arrange in alphabetical order.We must apologize for our carelessness about this point.
We have arranged all keywords in alphabetical order.
(4) Statistical analysis: You didn’t compare the significant difference between treatment means by any method such as Duncan’s test or Tukey? Why?Table 1 and Table 2: Please compare the significant difference between treatment means and show difference with letters.
We have combined Table 1 and Table 2 to one Table 1 and the significant difference between treatment means by methods of Duncan’s test has been marked by different letters in Table 1.
(5) Table 4 → use journal format (Remove left, right and inside borders)We must apologize for our carelessness about this point.
The format of this table has been corrected.
(6) Conclusions: this section given here are do not reflect what had been achieved including many speculations. Hence these need to be suitably modified.
We have modified the conclusion section probably.
Reviewer 2 Report
Interesting work.
I suggest the following for further improvement of the MS.
1. Please expand the figure legends and table headings so that an author can understand the results more clearly without going back to the methods for clarifications.
2. A line explaining the retrogradation process introduction will make it more clear to readers not familiar with this process.
Author Response
Review #2:Comments and Suggestions for Authors
Interesting work.I suggest the following for further improvement of the MS.
1. Please expand the figure legends and table headings so that an author can understand the results more clearly without going back to the methods for clarifications.
We have modified the figure legends and table headings probably.
2. A line explaining the retrogradation process introduction will make it more clear to readers not familiar with this process.
One sentence of “The starch retrogradation is essentially the repolymerization of dispersed amylopectin into regular structure by hydrogen bond” has been inserted into introduction section.
Reviewer 3 Report
Authors studied five different source of amylopectins from starches with alkali soluble glutenin on disulfide bond formation before and after retrogradation. They used a range of techniques to show the molecular weight and chain length distribution profiles, FTIR and NMR spectroscopy, and X-ray analysis. They showed that wheat amylopectin promoted the formation of disulfide bonds. The work is a novel study, but there is a suggestion to improve it:
- Authors used T-test for comparisons of tow samples. Its recommended to use analysis of variance and use a Dunnett’s test to compare the five treatment with control.
Author Response
Authors studied five different source of amylopectins from starches with alkali soluble glutenin on disulfide bond formation before and after retrogradation. They used a range of techniques to show the molecular weight and chain length distribution profiles, FTIR and NMR spectroscopy, and X-ray analysis. They showed that wheat amylopectin promoted the formation of disulfide bonds. The work is a novel study, but there is a suggestion to improve it:
Authors used T-test for comparisons of tow samples. Its recommended to use analysis of variance and use a Dunnett’s test to compare the five treatment with control.
The significant differences between means of five treated samples have been calculated using a Dunnett’s test and marked by different letters in Table 1.
Reviewer 4 Report
The manuscript by Zhou et al. assessed molecular changes in model systems containing alkali-soluble glutenin and amylopectin isolated from five botanical sources. The amylopectin as well as the model systems were characterized by various methods. The main objective of the study was to evaluate the formation of protein cross-links via disulfide bond formation in the presence of the tested starches, in fresh as well as retrograded samples. Based on the results, a mechanism to explain differences among the starches was postulated.
Strengths of the study include that multiple methods were selected to characterize the systems. However, there are some weaknesses that need to be addressed before publication, and certain questions require clarification.
At times the text is hard to read, and I have made several suggestions below. However, my main concern is that the authors propose that hydroxyprolin is covalently linked to glucose units in amylopectin, and this requires more elaboration. What exact kind of linkage would this be, which position in the glucose unit is affected? The mechanism proposed in Figure 4 seems somewhat far-fetched. It would involve the formation of an alkyl-aryl ether by mere mixing - usually such a compound can only be synthesized at relatively high temperatures. Moreover, in the next step, the aromaticity of the ring would be lost due to addition of the S-S-R group - again, this is not very plausible to happen without severe conditions. Finally, the ether bond would need to be cleaved, but even though the ring is not aromatic any more, no tautomerization to the keto group would occur? What would cause the hydrolysis of the ether bond during retrogradation? Also, why would it be only hydoxyprolin that undergoes such reactions with amylopectin, and no other amino acid? Is there any literature that has indicated that amylopectin specifically binds to hydroxyproline? The authors have not discussed where the hydroxyproline would have come from. There is literature on the hydroxyproline content in wheat arabinogalactans, would that be the proposed source?
Moreover, the authors write that amylopectin "may activate" enzymes, but why and how exactly would it do that? Also, where would these enzymes come from? The authors write that glutenin was prepared from gluten (L138). I could not find it listed where the gluten was from, the materials section 2.1 does not state it (it should be included). But if it was gluten, then it would likely have been prepared by washing out the starch. Would such a process not remove most albumins and globulins too? Enzymes would thus be lost in such a process, so I don't see how they could catalyze any reactions in this system. If the authors think that enzymes were still present in the systems, then this should be assessed and shown. There was an autoclaving treatment, for 30 min. Would this not induce changes that are different than by just mixing or retrograding. Also, even if there were enzymes present, would they not have been denatured?
It was also unclear to me why H bonds between amylopectin and C=O in prolin of ASG would stabilize helices (L574). Prolin is a helix breaker, why would it stabilize helices?
Was it ever tested if the isolated amylopectins contained protein by either Dumas or Kjeldahl? It would be good to know the N content of the materials.
Thus, overall, there is more explanation and incorporation of literature needed to support the interpretation and hypothesis. The authors should also propose how their hypothesized mechanism can be tested, because from an organic chemistry perspective it is not very likely.
Below are some editorial comments:
L11: By writing "the dough", one indicates that a very specific dough system is being referred to. However, I think this sentence is meant as a general remark. So I would remove "the", and maybe write "dough systems" instead.
L12: Is the problem maybe inconsistency? To me, it would sound better to write that the effects have been inconsistent. I recommend "However, inconsistent effects of these starches on dough quality have been reported, especially in refrigerated dough."
L14: To cut words, I recommend removing "The results showed that". The sentence can start with "Both what and cassava ..." and all the relevant information is preserved.
L18/19: This is a long sentence and the transition between the two parts is not quite clear. I recommend to split it into 2 sentences.
L20: It is unclear what the importance of hydroxyproline is and how it relates to disulfide bond formation and amylopectin. I suggest that some explanation is added to abstract to make it clearer, especially since in the last sentence it is proposed to use it as a quality marker in breeding.
L20: Insert "for" before "cassava"
L24/25: I would change the sentence to "A possible pathway of disulfide bond formation in ASGs by amylopectin addition is proposed."
L30: Is it important that China is self sufficient in wheat? It is unclear why the phrase was included
L31: "wheaten food" sounds a bit strange
L32: A review would be better as reference 1
L32: Gliadins can also be extracted with aqueous propanol, so I suggest replacing "ethanol" by "alcohol" or "short-chain alcohol" in this sentence
L46: If author names are part of the sentences, shouldn't the year be included? This is common practice, and the authors have in fact done it elsewhere in the manuscript too. The comma after Gianibelli is also not needed.
L58: I assume that several of the authors have made this finding about wheat amylopectin and SS-formation. So then it should be " authors' " and not "author's" because the latter would only refer to one person, not several.
L60: A general comment: There are several times that unpublished results are mentioned. I would write "In pre-trials" or "in unpublished previous studies" or something like that. Have any other authors reported on such a finding?
L66: use the past tense here, "were" instead of "are"
L69: I don't think that there should be a comma after "et al."
L71: H bonds are by definition not-covalent, so the term can be removed from the sentence
L71/72: It is not clear what "superpose the three-dimensional elastic network" means
L72/73: This sentence is quite confusing, please rephrase
L74: again, use past tense, not "is"
L78-79: I do not think that results need to be stated here in the introduction already. The part of "and it was found ..." can be removed from the sentence. Then "is to ascertain this phenomenon" should be rephrased
L81: It is called solid-state, not solid 13C NMR. The authors are using the correct name in other places of the manuscript (e.g., L239), but not everywhere. This should be corrected.
L82: The sentence "In actual production ..." seems inappropriate for an introduction. Moreover, there is a very similar sentence in the conclusion too.
L105: "according to basic theory in the literature" sounded a bit odd.
L106: resolve or dissolve (solubilize)?
L106: "wipe out" is not a very technical expression
L123: lipase and protease would not really hydrolyze amylopectin, would they? They would hydrolyze co-extracted or associated compounds, but not glycosidic linkages. This phrase is used several times, please change it
L138: "by modified method in reference" does not sound correct
L150: "was motionlessly placed in a beaker" - as soon as something is "placed in a beaker" it is moved, so I don't think it can be done "motionlessly"
L238: Why is the header named 2.2.913. C? This seems like a copy-paste error from some other text
L250: What does "two-sample for variances of t-test" mean? Also, there were several samples that were compared to each other. Why not do an ANOVA and post-hoc analysis? Moreover, the factor same sample before and after retrogradation could have also been compared with a t-test. So I think that the statistical analysis should be modified. Then Table 1 and 2 could also be combined.
L254: The grammar in this sentence is off
L259: I think "pack" would be better than "package".
L264: Was this "other project" never published?
L264: "will make the dough out of shape" does not sound grammatically correct
Table 1 & 2: It is very surprising that mung bean or sweet potato starch were not found different to the control, because the difference between means is quite large, and for mung bean the standard deviation is not so large. What were the p-values? In table 2, Cassava starch was not different to the control, even though one has a mean of 0.03 +/- 0.00 the other one of 0.26 +/- 0.05? In contrast, maize had a value of 0.16 +/- 0.03 and it was found to be different? I see that the standard deviation was lower for maize than for cassava, but the difference between means is 0.13 vs. 0.23. How can this be, what were the p-values?
L330: Instead of "will be found in", maybe write "This will be further discussed in sections XX below"
L333: Some of the tables (4 & 5) are not formatted very well, all the lines are currently still visible.
L345: What is the "smooth degree of the region"? Similarly, what is "smooth change of the field" (L356, L357)?
L364: "should" some a bit patronizing, maybe write "were"
L367: on the one hand, the authors write "demonstrates", a verb that implies a high degree of certainty. On the other hand, "might" is used, which is more careful. So this sounds a bit odd and I would remove or modify one of these words
L394: What does "interact firmly with other keys" mean?
L397: It is not clear why the MW would play such a role.
L400: Why is this numbered "3.413. C"?
L404: The grammar is off
L401 & 405: Use past tense
L455: "the enhance of" - the increase?
L493: "ignorable" is maybe not the best word choice. "nd" for not detected? Or traces?
L504: "unravel the essence for the addition" does not sound correct
L508: "Dimer glutathione reductase in sample distribution inhomogeneity" sounds grammatically odd, but it is also unclear what data this is based on because the activity or even just presence of this enzyme have not been assessed.
L547: "The possible way" = "Mechanism"?
L562: Please elaborate on what ref 55 reported and how it is aligned with the current study
L564: In what form would the SH groups "dissociate" from cysteine, and what would happen with the residual cysteine?
L574: "which is puled by helix" - unclear statement
L585: "It's easy" is not a very technical expression. Also, avoid contractions (such as "it's", "don't" etc.) in technical writing.
Author Response
Reviewer 4:
Comments and Suggestions for Authors
The manuscript by Zhou et al. assessed molecular changes in model systems containing alkali-soluble glutenin and amylopectin isolated from five botanical sources. The amylopectin as well as the model systems were characterized by various methods. The main objective of the study was to evaluate the formation of protein cross-links via disulfide bond formation in the presence of the tested starches, in fresh as well as retrograded samples. Based on the results, a mechanism to explain differences among the starches was postulated.
Strengths of the study include that multiple methods were selected to characterize the systems. However, there are some weaknesses that need to be addressed before publication, and certain questions require clarification.
At times the text is hard to read, and I have made several suggestions below. However, my main concern is that the authors propose that hydroxyprolin is covalently linked to glucose units in amylopectin, and this requires more elaboration. What exact kind of linkage would this be, which position in the glucose unit is affected? The mechanism proposed in Figure 4 seems somewhat far-fetched. It would involve the formation of an alkyl-aryl ether by mere mixing - usually such a compound can only be synthesized at relatively high temperatures. Moreover, in the next step, the aromaticity of the ring would be lost due to addition of the S-S-R group - again, this is not very plausible to happen without severe conditions. Finally, the ether bond would need to be cleaved, but even though the ring is not aromatic any more, no tautomerization to the keto group would occur? What would cause the hydrolysis of the ether bond during retrogradation? Also, why would it be only hydoxyprolin that undergoes such reactions with amylopectin, and no other amino acid? Is there any literature that has indicated that amylopectin specifically binds to hydroxyproline? The authors have not discussed where the hydroxyproline would have come from. There is literature on the hydroxyproline content in wheat arabinogalactans, would that be the proposed source?
Moreover, the authors write that amylopectin "may activate" enzymes, but why and how exactly would it do that? Also, where would these enzymes come from? The authors write that glutenin was prepared from gluten (L138). I could not find it listed where the gluten was from, the materials section 2.1 does not state it (it should be included). But if it was gluten, then it would likely have been prepared by washing out the starch. Would such a process not remove most albumins and globulins too? Enzymes would thus be lost in such a process, so I don't see how they could catalyze any reactions in this system. If the authors think that enzymes were still present in the systems, then this should be assessed and shown. There was an autoclaving treatment, for 30 min. Would this not induce changes that are different than by just mixing or retrograding. Also, even if there were enzymes present, would they not have been denatured?
We thank the reviewer #4 very much for questioning the reaction mechanism in Figure 4, surely, such deduction is far-fetched. These assumptions are based on an analysis of the results of nuclear magnetic carbon spectroscopy in Fig. 2 and Table 5 (in modified paper) and reference, if there is something wrong with the assignments of those carbons, we are very open to different opinions. I am sure there are many chemical and biological reactions that we don't yet understand in formation of dough. Whether certain enzymes exist needs further study to verified. If reviewers could offer a more reasonable analysis, we would appreciate them and accept.
It was also unclear to me why H bonds between amylopectin and C=O in proline of ASG would stabilize helices (L574). Proline is a helix breaker, why would it stabilize helices?
We thank the reviewer #4 very much for the analysis and we replace proline by glutamine, because they are the two main amino acids in alkali-soluble glutenin.
Was it ever tested if the isolated amylopectins contained protein by either Dumas or Kjeldahl? It would be good to know the N content of the materials.
We have determined N contents of amylopectins in other experiments by Kjeldahl methods, they are about 0.01%~0.1%. Those proteins are very difficult to be isolated.
Thus, overall, there is more explanation and incorporation of literature needed to support the interpretation and hypothesis. The authors should also propose how their hypothesized mechanism can be tested, because from an organic chemistry perspective it is not very likely.
We regard that process of dough formation is a biological and chemical action in which there's a lot of organic and inorganic stuff involved. We will design more experiments to uncover them step by step according to advice of reviewer # 4.
Below are some editorial comments:
L11: By writing "the dough", one indicates that a very specific dough system is being referred to. However, I think this sentence is meant as a general remark. So I would remove "the", and maybe write "dough systems" instead.
We have revised them according to advice of reviewer.
L12: Is the problem maybe inconsistency? To me, it would sound better to write that the effects have been inconsistent. I recommend "However, inconsistent effects of these starches on dough quality have been reported, especially in refrigerated dough."
We have revised them according to advice of reviewer.
L14: To cut words, I recommend removing "The results showed that". The sentence can start with "Both what and cassava ..." and all the relevant information is preserved.
We have revised them according to advice of reviewer.
L18/19: This is a long sentence and the transition between the two parts is not quite clear. I recommend to split it into 2 sentences.
We have revised them according to advice of reviewer.
L20: It is unclear what the importance of hydroxyproline is and how it relates to disulfide bond formation and amylopectin. I suggest that some explanation is added to abstract to make it clearer, especially since in the last sentence it is proposed to use it as a quality marker in breeding.
We have revised them according to advice of reviewer.
L20: Insert "for" before "cassava"
We have revised them according to advice of reviewer.
L24/25: I would change the sentence to "A possible pathway of disulfide bond formation in ASGs by amylopectin addition is proposed."
We have revised them according to advice of reviewer.
L30: Is it important that China is self sufficient in wheat? It is unclear why the phrase was included
We have deleted the words of “self sufficient”.
L31: "wheaten food" sounds a bit strange
They have been replaced by “ wheat food”.
L32: A review would be better as reference 1
We have revised them according to advice of reviewer.
L32: Gliadins can also be extracted with aqueous propanol, so I suggest replacing "ethanol" by "alcohol" or "short-chain alcohol" in this sentence
We have revised them according to advice of reviewer.
L46: If author names are part of the sentences, shouldn't the year be included? This is common practice, and the authors have in fact done it elsewhere in the manuscript too. The comma after Gianibelli is also not needed.
We have revised them according to advice of reviewer.
L58: I assume that several of the authors have made this finding about wheat amylopectin and SS-formation. So then it should be " authors' " and not "author's" because the latter would only refer to one person, not several.
We have revised them according to advice of reviewer.
L60: A general comment: There are several times that unpublished results are mentioned. I would write "In pre-trials" or "in unpublished previous studies" or something like that. Have any other authors reported on such a finding?
We have revised them according to advice of reviewer.
L66: use the past tense here, "were" instead of "are"
We have revised them according to advice of reviewer.
L69: I don't think that there should be a comma after "et al."
We have revised them according to advice of reviewer.
L71: H bonds are by definition not-covalent, so the term can be removed from the sentence
Not-covalent has been deleted in the sentence.
L71/72: It is not clear what "superpose the three-dimensional elastic network" means
L72/73: This sentence is quite confusing, please rephrase
We have modified the sentence to “The starch retrogradation is essentially the repolymerization of dispersed amylopectin into regular structure by hydrogen bond. Hydrogen bonds produced in starch retrogradation also probably prevented the formation of three-dimensional elastic network of alkali soluble glutenin [12], thus brought certain effects on disulfide bond formation.”
L74: again, use past tense, not "is"
We have revised them according to advice of reviewer.
L78-79: I do not think that results need to be stated here in the introduction already. The part of "and it was found ..." can be removed from the sentence. Then "is to ascertain this phenomenon" should be rephrased
We have revised them according to advice of reviewer.
L81: It is called solid-state, not solid 13C NMR. The authors are using the correct name in other places of the manuscript (e.g., L239), but not everywhere. This should be corrected.
We have revised them according to advice of reviewer.
L82: The sentence "In actual production ..." seems inappropriate for an introduction. Moreover, there is a very similar sentence in the conclusion too.
We have revised them according to advice of reviewer.
L105: "according to basic theory in the literature" sounded a bit odd.
We have replaced “theory” by “principle”.
L106: resolve or dissolve (solubilize)?
It should be “dissolve”, we have changed it.
L106: "wipe out" is not a very technical expression
Those words have been replaced by “remove”.
L123: lipase and protease would not really hydrolyze amylopectin, would they? They would hydrolyze co-extracted or associated compounds, but not glycosidic linkages. This phrase is used several times, please change it
We have revised them according to advice of reviewer.
L138: "by modified method in reference" does not sound correct
Those words have been changed to “method in reference with certain modification”.
L150: "was motionlessly placed in a beaker" - as soon as something is "placed in a beaker" it is moved, so I don't think it can be done "motionlessly"
The word of “motionlessly” has been changed to “gently”.
L238: Why is the header named 2.2.913. C? This seems like a copy-paste error from some other text
There should be a space between 2.2.9 and 13C, we have changed it. It occurs during file conversion.
L250: What does "two-sample for variances of t-test" mean? Also, there were several samples that were compared to each other. Why not do an ANOVA and post-hoc analysis? Moreover, the factor same sample before and after retrogradation could have also been compared with a t-test. So I think that the statistical analysis should be modified. Then Table 1 and 2 could also be combined.
We have revised them according to advice of reviewer.
L254: The grammar in this sentence is off
The sentence has been changed to “The greater disulfide bonds content brought more stable to the doughs' network structure”.
L259: I think "pack" would be better than "package".
We have revised them according to advice of reviewer.
L264: Was this "other project" never published?
Yes, it is unpublished results, we have marked it.
L264: "will make the dough out of shape" does not sound grammatically correct
The sentence has been changed to “…which showed that the dough would not be formed when 20% retrograded maize amylose were added in…”
Table 1 & 2: It is very surprising that mung bean or sweet potato starch were not found different to the control, because the difference between means is quite large, and for mung bean the standard deviation is not so large. What were the p-values? In table 2, Cassava starch was not different to the control, even though one has a mean of 0.03 +/- 0.00 the other one of 0.26 +/- 0.05? In contrast, maize had a value of 0.16 +/- 0.03 and it was found to be different? I see that the standard deviation was lower for maize than for cassava, but the difference between means is 0.13 vs. 0.23. How can this be, what were the p-values?
Such results are attributed to the facts that there is a big difference among the repeated values of those samples, although we have repeated the experiments for several times. The formation of disulfide bonds in these samples is significantly affected by uncontrolled environmental conditions such as sample uniformity, consistency of stirring degree, etc.
L330: Instead of "will be found in", maybe write "This will be further discussed in sections XX below"
We have revised them according to advice of reviewer.
L333: Some of the tables (4 & 5) are not formatted very well, all the lines are currently still visible.
We have revised them according to advice of reviewer.
L345: What is the "smooth degree of the region"? Similarly, what is "smooth change of the field" (L356, L357)?
The meaning of smooth degree in here is the smoothness and sharpness of the infrared absorption peak, more smooth degree corresponds to more hydrogen bonds. We have marked them in the sentence at suitable position.
L364: "should" some a bit patronizing, maybe write "were"
We have revised them according to advice of reviewer.
L367: on the one hand, the authors write "demonstrates", a verb that implies a high degree of certainty. On the other hand, "might" is used, which is more careful. So this sounds a bit odd and I would remove or modify one of these words
The word of “might” in the sentence has been deleted.
L394: What does "interact firmly with other keys" mean?
We have changed the sentence by “…disulfide bonds are vibrationally bound…”
L397: It is not clear why the MW would play such a role.
This is a good question, and we have thought it for a long time. We regard that as ASG is combined to amylopectin with higher molecular weight, the spatial structure of ASG molecules can be changed and stabilized under the action of water molecules, so that the structure is conducive to the formation of disulfide bonds.
L400: Why is this numbered "3.413. C"?
There should be a space between 3.4 and 13C, we have changed it. It occurs during file conversion.
L404: The grammar is off
We have revised them according to advice of reviewer.
L401 & 405: Use past tense
We have revised them according to advice of reviewer.
L455: "the enhance of" - the increase?
We have revised them according to advice of reviewer.
L493: "ignorable" is maybe not the best word choice. "nd" for not detected? Or traces?
We have revised them according to advice of reviewer.
L504: "unravel the essence for the addition" does not sound correct
The sentence has been changed to “Those findings for mung bean and sweet potato amylopectin and ASG complexes before and after retrogradation shed light on the nature of the effects of the addition of mung bean and sweet potato starches on dough properties”
L508: "Dimer glutathione reductase in sample distribution inhomogeneity" sounds grammatically odd, but it is also unclear what data this is based on because the activity or even just presence of this enzyme have not been assessed.
This statement is speculative and we add a word “might” in it.
L547: "The possible way" = "Mechanism"?
They are not the same, we have deleted the word “mechanism” in the paper.
L562: Please elaborate on what ref 55 reported and how it is aligned with the current study
One sentence of “, suggesting that the proportion of α-helices, β-turns, and antiparallel β-sheets increases when glutenin is mixed with starch at low temperature” has been inserted in.
L564: In what form would the SH groups "dissociate" from cysteine, and what would happen with the residual cysteine?
This description is hypothetical and has no scientific basis.
L574: "which is puled by helix" - unclear statement
We have revised them according to advice of reviewer.
L585: "It's easy" is not a very technical expression. Also, avoid contractions (such as "it's", "don't" etc.) in technical writing.
We have revised them according to advice of reviewer.
Reviewer 5 Report
This manuscript look into "Effects of amylopectins from five different sources on disulfide bond formation in alkali-soluble glutenin".
The introduction is comprehensive. The methodology is also extensive to answer the research question. This study has resulted in convincing results on the effect of amylopectin on the disulfide bond formation in alkali-soluble glutenin.
Author Response
This manuscript looks into "Effects of amylopectins from five different sources on disulfide bond formation in alkali-soluble glutenin".
The introduction is comprehensive. The methodology is also extensive to answer the research question. This study has resulted in convincing results on the effect of amylopectin on the disulfide bond formation in alkali-soluble glutenin.
We have modified grammar of repetitive sentence in all paper.
Round 2
Reviewer 2 Report
Accept for publication
Author Response
Response to Reviewer 2 Comments
Point 1: English language and style are fine/minor spell check required.
Response 1: Please provide your response for Point 1. (in red)
- in line 12, ”in“ has been changed to “of”. In line 14,”it “ has been changed to “ASG” to make the meaning clearer.
- In line 15,”above “ has been changed to “above-mentioned”. In line 16, “by” has been changed to “through”; in line 17, “analysis” has been added after “…XRD”.
- In line 22,”Presence of hydroxyproline only “ has been changed to “Hydroxyproline only exixted”.
- In line 67,”investigated “ has been changed to “investigate”.
- In line 68,”Corresponding “ has been changed to “corresponding”.
- In line 71,”whether “ has been changed to “but whether”.
- In line 89,” economic and trade company “ has been changed to “Economic and Trade Company”.
- In line 102,” are “ has been changed to “were”.
- In line 103,” German “ has been changed to “Germany”.
- In line 107,” Principle “ has been changed to “Principles”.
- In line 108,” from which “ has been inserted before “amylose…”.
- In line 120,” were “ has been changed to “was”.
- In line 121,” Repeat “ has been changed to “Repeatedly”.
- In line 127,” according “ has been changed to “according to the”.
- In line 132,” At last “ has been changed to “Finally”.
- In line 138,” have “ has been changed to “had”. In line 149, “on” has been changed to “no”.
- In line 197,” were “ has been changed to “was”.
- In line 205,” equipped “ has been changed to “was equipped”.
- In line 209,” was “ has been changed to “were”.
- In line 257,” stable “ has been changed to “stable state”.
- In line 265,” others “ has been changed to “other”.
- In line 267,” what “ has been changed to “as”.
- In line 268,” which “ has been deleted.
- In line 267,” what “ has been changed to “as”.
- In line 269,” were“ has been changed to “was”.
- In line 271-272,” convince “ and “promote” have been changed to “which convinced” and “promoted”, respectively.
- In line 272,” show “ has been changed to “shows”.
- In line 274,” is “ has been changed to “are”.
- In line 275,” that “ has been changed to “those”.
- In line 312,” they “ has been changed to “which”.
- In line 314,” can “ has been inserted before “…also…”.
- In line 316,” do “ has been changed to “does”.
- In line 328,” appearance “ has been changed to “and the appearance”.
- In line 338,” are “ has been changed to “is”.
- In line 345,” and “ has been inserted before “our results…”.
- In line 347,”regard “ has been changed to “believe”.
- In line 347,”were “ has been changed to “are”.
- In line 371,” and “ has been inserted before “vice versa…”.
- In line 372,”Comparing“ has been changed to “Compared”.
- In line 384,”to“ has been changed to “from”.
- In line 388,”a“ has been changed to “an”.
- In line 399,”Marked“ has been changed to “marked”.
- In line 442-445,” When ASG is mixed with wheat amylopectin in Fig.2a (blue line) and Table 5, a process to enhance formation of disulfide bond, resonances for Qδ and C=O shift to lower and higher field, respectively, and those of Yγ (131.4 ppm) in ASG and HYP Cβ (31.5 ppm) in wheat amylopectin disappear “ has been changed to “When ASG is mixed with wheat amylopectin in Fig.2a (blue line) and Table 5, the formation of disulfide bond enhances. During the mixed process, the resonances for Qδ and C=O shift to lower and higher field, respectively, and those of Yγ (131.4 ppm) in ASG and HYP Cβ (31.5 ppm) in wheat amylopectin disappear”.
- In line 469-471,” Compared with wheat amylopectin, maize amylopectin also combines with protein containing Gln, but absence of resonance at 31.5 ppm for HYP Cγ in Fig. 2 b and Table 5. “ has been changed to “Compared with wheat amylopectin, maize amylopectin also combines with protein containing Gln, but no resonance at 31.5 ppm for HYP Cγ in Fig. 2 b and Table 5 appears.”.
- In line 471-472,” When it is mixed with ASG, the difference is present of resonance at 131.7 ppm for Yγ and absence of resonance at 132.6 ppm for Yδ, “ has been changed to “When it is mixed with ASG, the difference of resonance is present at 131.7 ppm for Yγ and absence of resonance at 132.6 ppm for Yδ,”.
- In line 472,”and“ has been changed to “but”.
- In line 480,”with“ has been changed to “from”.
- In line 480-483,” It is worth noting that there is no trace of protein in cassava amylopectin in Fig. 2 c (marked with dashed arrows) and Table 5, this may be the fundamental reason why it cannot promote the formation of a large number of glutenin disulfide bonds during dough formation. “ has been changed to “It is worth noting that there is no trace of protein in cassava amylopectin in Fig. 2 c (marked with dashed arrows) and Table 5. This may be the fundamental reason why it cannot promote the formation of a large number of glutenin disulfide bonds during dough formation”.
- In line 483-486,” When cassava amylopectin is mixed with ASG, as shown in Fig. 2 c and Table 5, all resonances for Tyr at 132.6/131.4/128.5 ppm disappear, they are probably buried in new secondary structures of intra-molecular aggregation extended β-sheet and β-turn “ has been changed to “When cassava amylopectin is mixed with ASG, as shown in Fig. 2 c and Table 5, all resonances for Tyr at 132.6/131.4/128.5 ppm disappear. They are probably buried in new secondary structures of intra-molecular aggregation extended β-sheet and β-turn”.
- In line 493,” respectively “ has been changed to “as is shown respectively”.
- In line 496,” ones “ has been changed to “one”.
- In line 505,” No matter “ has been changed to “Whether”.
- In line 511,” imply “ has been changed to “implies”.
- In line 521,” contains “ has been changed to “containing”.
- In line 530,” show “ has been changed to “shows”.
- In line 539,” to “ has been changed to “as”.
- In line 547,” to “ has been changed to “as”.
- In line 548,” as “ has been inserted before “mixture…”.
- In line 549,” make “ has been changed to “makes”.
- In line 566,” promotes “ has been changed to “promote”.
- In line 626,” , “ has been inserted before “and”.
- In line 627,” being “ has been deleted.
- In line 633,” graduates Zhixiang He in the 21th Speciality “ has been changed to “Zhixiang He, one of our graduates, in the 21st Speciality”.
Reviewer 4 Report
The authors have substantially revised their manuscript, and it is much clearer now.
However, there is one issue that prevents me from accepting it in its present form: Several reviewers (including myself) have stated that the multiple t-tests are not appropriate. Reviewers had been very clear that the treatments should be compared with the control. While the authors have done this for the starch samples, they have not included the control. This needs to be changed! That way, it can be discussed which samples are statistically different from the control or from each other.
There are still some editorial issues, as pointed out below - but these are minor comments. Overall, the new text is quite good!
L58: I recommend writing "wheat-based food" instead of "wheat food".
L104: I'd remove "the" before "starch"
L105: I would write "a more ordered" instead of "regular". Is this really a repolymerization if it only involves non-covalent bonds? I would write something like "Starch retrogradation essentially involves a reassociation of dispersed amylopectin into a more ordered structure stabilized by hydrogen bonds".
L116: Use past tense - "was", not "is"
L432: I would replace "symbol" by "indicator"
L446: "combined with", not "combined to"
L666: "in a way of glutation disulfide" sounds odd. Is it meant that disulfides between glutathione and the non-carbohydrate part of these amylopectins?
L447: I'd write "may", not "can"; and "by" instead "under the action of"
Author Response
Reviewer #4:
Comments and Suggestions for Authors
The authors have substantially revised their manuscript, and it is much clearer now.
However, there is one issue that prevents me from accepting it in its present form: Several reviewers (including myself) have stated that the multiple t-tests are not appropriate. Reviewers had been very clear that the treatments should be compared with the control. While the authors have done this for the starch samples, they have not included the control. This needs to be changed! That way, it can be discussed which samples are statistically different from the control or from each other.
We apologize for our carelessness about this point, we have modified them now.
There are still some editorial issues, as pointed out below - but these are minor comments. Overall, the new text is quite good!
L58: I recommend writing "wheat-based food" instead of "wheat food".
We have modified it according to advice of reviewer #4.
L104: I'd remove "the" before "starch"
We have modified it according to advice of reviewer #4.
L105: I would write "a more ordered" instead of "regular". Is this really a repolymerization if it only involves non-covalent bonds? I would write something like "Starch retrogradation essentially involves a reassociation of dispersed amylopectin into a more ordered structure stabilized by hydrogen bonds".
We have modified it according to advice of reviewer #4. We learn more from this point, thank you very much!
L116: Use past tense - "was", not "is"
We have modified it according to advice of reviewer #4.
L432: I would replace "symbol" by "indicator"
We have modified it according to advice of reviewer #4.
L446: "combined with", not "combined to"
We have modified it according to advice of reviewer #4.
L666: "in a way of glutation disulfide" sounds odd. Is it meant that disulfides between glutathione and the non-carbohydrate part of these amylopectins?
Here, glutathione disulfide means glutathione polymer linked by disulfide bonds. We have changed it.
L447: I'd write "may", not "can"; and "by" instead "under the action of"
We have modified it according to advice of reviewer #4.
Again, we thank the reviewer #4 very much for those editorial comments, we accept all of them. They are so valuable to us, and you are rising us up, you are a nice person, thank thank thank you very much!!! God bless you! Happy New Year!